# SnoRNAs and miRNAs Networks Underlying COVID-19 Disease Severity

**DOI:** 10.3390/vaccines9101056

**Published:** 2021-09-23

**Authors:** Aijaz Parray, Fayaz Ahmad Mir, Asmma Doudin, Ahmad Iskandarani, Ibn Mohammed Masud Danjuma, Rahim Ayadathil Thazhhe Kuni, Alaaedin Abdelmajid, Ibrahim Abdelhafez, Rida Arif, Mohammad Mulhim, Mohammad Abukhattab, Shoukat Rashhid Dar, Ala-Eddin Al Moustafa, Eyad Elkord, Abdul Latif Al Khal, Abdel-Naser Elzouki, Farhan Cyprian

**Affiliations:** 1The Neuroscience Institute, Academic Health System, Hamad Medical Corporation, Doha 3050, Qatar; aparray@hamad.qa (A.P.); RKuni1@hamad.qa (R.A.T.K.); 2Qatar Metabolic Institute, Academic Health System, Hamad Medical Corporation, Doha 3050, Qatar; FMir1@hamad.qa (F.A.M.); AIskandarani@hamad.qa (A.I.); 3Department of Math and Science, Community College of Qatar, Doha 7344, Qatar; asmma.doudin@ccq.edu.qa; 4College of Medicine, QU Health, Qatar University, Doha 2713, Qatar; MDanjuma@hamad.qa (I.M.M.D.); iabdelhafez@hamad.qa (I.A.); ridaarif0@gmail.com (R.A.); aalmoustafa@qu.edu.qa (A.-E.A.M.); AElzouki@hamad.qa (A.-N.E.); 5Internal Medicine Department, Hamad General Hospital, Hamad Medical Corporation, Doha 3050, Qatar; 6Communicable Diseases Center, Hamad Medical Corporation, Doha 3050, Qatar; aabdelmajid@hamad.qa (A.A.); msaleh@hamad.qa (M.M.); MABUKHATTAB@hamad.qa (M.A.); aalkhal@hamad.qa (A.L.A.K.); 7Department of Neurosurgery, Hamad Medical Corporation, Doha 3050, Qatar; 8Department of Emergency, Hamad Medical Corporation, Doha 3050, Qatar; Sdar1@hamad.qa; 9Biomedical and Pharmaceutical Research Unit, QU Health, Qatar University, Doha 2713, Qatar; 10Biomedical Research Centre, QU Health, Qatar University, Doha 2713, Qatar; 11Natural and Medical Sciences Research Centre, University of Nizwa, Nizwa 208, Oman; E.Elkord@salford.ac.uk; 12Biomedical Research Centre, School of Science, Engineering and Environment, University of Salford, Manchester M5 4BR, UK

**Keywords:** snoRNA, miRNA, SARS-CoV-2, COVID-19, biomarkers

## Abstract

There is a lack of predictive markers for early and rapid identification of disease progression in COVID-19 patients. Our study aims at identifying microRNAs (miRNAs)/small nucleolar RNAs (snoRNAs) as potential biomarkers of COVID-19 severity. Using differential expression analysis of microarray data (n = 29), we identified hsa-miR-1246, ACA40, hsa-miR-4532, hsa-miR-145-5p, and ACA18 as the top five differentially expressed transcripts in severe versus asymptomatic, and ACA40, hsa-miR-3609, ENSG00000212378 (SNORD78), hsa-miR-1231, hsa-miR-885-3p as the most significant five in severe versus mild cases. Moreover, we found that white blood cell (WBC) count, absolute neutrophil count (ANC), neutrophil (%), lymphocyte (%), red blood cell (RBC) count, hemoglobin, hematocrit, D-Dimer, and albumin are significantly correlated with the identified differentially expressed miRNAs and snoRNAs. We report a unique miRNA and snoRNA profile that is associated with a higher risk of severity in a cohort of SARS-CoV-2 infected patients. Altogether, we present a differential expression analysis of COVID-19-associated microRNA (miRNA)/small nucleolar RNA (snoRNA) signature, highlighting their importance in SARS-CoV-2 infection.

## 1. Introduction

At the beginning of 2020, the World Health Organization (WHO) declared coronavirus disease-2019 (COVID-19) as a global pandemic. The causative organism, severe acute respiratory syndrome coronavirus 2 (SARS-CoV-2), exhibits a wide spectrum of clinical manifestations in disease-ridden patients. Differences in the severity of COVID-19 ranges from asymptomatic infections and mild cases to the severe form, leading to acute respiratory distress syndrome (ARDS) and multiorgan failure with poor survival [1]. Moreover, the mortality rate is influenced by aging, viral strain, pre-existing comorbidities, and the degree of immunocompromise. Indeed, the health and socioeconomic implications of the COVID-19 pandemic are enormous, and thus warrants the search for new interventions and treatment measures. Recent research has suggested that a unique non-coding RNA signature can aid in identifying the likelihood of developing specific disease outcomes [2]. Alterations of miRNA levels in the blood have been described in multiple inflammatory and infectious diseases, including SARS-related coronaviruses [3,4,5,6,7,8,9]. MiRNAs are endogenous small non-coding RNAs, around 22 nucleotides, that bind specific messenger RNAs (mRNAs) through complementary base-pairing [10]. Hence, miRNAs can regulate various cellular processes, including proliferation, apoptosis, and differentiation, by binding to the 3′UTR of target mRNAs inducing their degradation, thus serving a fundamental role in post-transcriptional repression [11,12]. In this context, a single miRNA can target several genes, and multiple miRNAs may regulate a single gene. Hence, identification of miRNAs as well as characterization of miRNA-mRNA interactions in SARS-CoV-2 infection is important to understand their role in disease pathogenesis, progression, and severity [13,14,15]. Accumulating evidence is also implicating snoRNAs in numerous physiological and pathological processes, including their interactions with some RNA viruses [16,17]. SnoRNAs have been shown to play a role in ribosomal RNAs (rRNAs) modification and maturation. These include pseudouridylation, 2′-O-methylation, polyadenylation, alternative splicing, and formation of protein complexes with fibrillarin that is a component of several ribonucleoproteins [18]. There are no reports yet with a comprehensive characterization of snoRNAs in SARS-CoV-2 infection. Here, we report for the first time the snoRNA signature, and similar to previous studies the miRNA signature in the peripheral blood of severe COVID-19 cases (n = 9), as compared to mild (n = 10) and asymptomatic (n = 10) patients. This study analyzed a total of 29 COVID-19 patients that were matched for age and comorbidities using Affymetrix GeneChip miRNA 4.0 array and were recruited between July and October 2020 in Qatar.

## 2. Materials and Methods

### 2.1. Study Design and Data Collection

This research is in accordance with the Reporting of Observational Studies in Epidemiology (STROBE) recommendations, the Code of Ethics of the World Medical Association. This study was granted ethical approval from the Medical Research Center at Hamad Medical Corporation (MRC-05-084, Immunological and immune-genetic investigations in COVID-19 patients with varying disease severity, 06/21/2020). All study participants gave written informed consent where possible, and deferred consent was obtained for ICU cases. From 173 recruited patients in this prospective cohort study, 29 male age-matched patients were included. All patients were previously diagnosed with COVID-19 using TaqPath COVID-19 Combo Kit (Thermo Fisher Scientific, Waltham, MA, USA), or Cobas SARS-CoV-2 Test (Roche Diagnostics, Rotkreuz, Switzerland), with a CT value < 30. Additional criterion for selection was age between 35 and 75 years. Participants were grouped into severe, mild, and asymptomatic. Classifying severe cases was based on the requirement of high-flow oxygen support and ICU admission (n = 9). Whereas mild patients were identified based on symptoms, and positive radiographic findings with pulmonary involvement (n = 10). Patients with no clinical presentation were labelled as asymptomatic cases (n = 10). In the severe group, three patients died with respiratory failure listed as the primary cause of death. Blood samples were collected in the PAXgene Blood RNA Tubes (PreAnalytiX) at the time of diagnosis, prior to isolation, or hospitalization. Routine laboratory tests that were performed include complete blood cell counts, electrolytes, glucose, albumin, total protein, C-reactive protein, procalcitonin, IL-6, D-dimers, ferritin, urea, and liver enzymes.

### 2.2. RNA Isolation and Quality Control

Peripheral venous blood (2.5 mL) in PaX gene tubes was inverted 8–10 times to ensure complete mixing with the lysis reagent. Tubes were then stored upright at room temperature for a minimum of 2 h, before transferring them to a freezer at −80 °C until RNA isolation. The RNA (both total and miRNA) was isolated with a blood miRNA kit from Qiagen (PreAnalytiX GmBH Hombrechtikon, Switzerland) following the manufacturer’s instructions. The concentrations and purity of the RNA samples were evaluated spectrophotometrically (Nanodrop ND-1000, Thermo, Wilmington, DE, USA). The RNA isolation process was validated by analyzing the integrity of RNA with the RNA 6000 Nano Chip Kit (Agilent, CA, USA); the presence of the small RNA fraction was confirmed by the Agilent Small RNA Kit (Agilent).

### 2.3. Microarray and Data Analysis

Affymetrix GeneChip miRNA 4.0 array was used following the manufacturer’s instructions for miRNA expression analysis. For each sample, 250 ng of RNA was labelled using the FlashTag™ Biotin RNA Labeling Kit (Genisphere, Hatfield, PA, USA). Following this, the labelled RNA was quantified, fractionated, and hybridized to the miRNA microarray with continuous agitation at 60 rpm for 16 h at 48 °C on a GeneChip Hybridization Oven 640. The miRNA microarray chips were then washed and stained using the GeneChip Fluidics Station 450 (Affymetrix, Santa Clara, CA, USA). Finally, the miRNA microarray chips were scanned using an Affymetrix GCS 3000 scanner (Affymetrix, Santa Clara, CA, USA), and the signal values were evaluated using the Affymetrix^®^ GeneChip™ Command Console software.

Raw data were extracted using the Affymetrix data extraction protocol in the Affymetrix GeneChip^®^ Command Console^®^ (AGCC) Software. All statistical analyses including raw data import, annotation, and quality control were conducted with R statistical software (version 4.0.4, R Foundation, Boston, MA, USA). R Bioconductor packages “oligo” [19], “oligoCalsses”, and “BioBase” were used for the pre-processing of microarray datasets [20]. Raw expression data in CEL files were read and parsed into ExpressionFeatureSet using the “read.celfiles” function, followed by sample and probe annotations with “AnnotatedDataFrame” function. Next background correction, normalization, and log_2_ transformation using the robust multiarray average (RMA) method was performed. The differentially expressed transcripts were screened out via the “limma” package [21]. The analysis was performed using the design matrix “~severity” as a factor, where it was either “asymptomatic”, “mild” or “severe”. A comparative analysis between different severity groups was carried out using thresholds of |log_2_FC| > 1.5 and FDR < 0.05. Hierarchical clustering of differentially expressed miRNAs was carried out using the “pheatmap” R package. The prediction of miRNA-target genes was performed using the “multiMiR” R package filtering with validated databases, including miRecords, miRTarBase, and TarBase, and the list of 26 differentially expressed miRNAs in severe vs asymptomatic and severe vs mild comparisons [22]. The package “miRBaseConverter” was used to retrieve the miRNA sequence based on the accession number [23]. The database miRTarBase was used with a threshold of 2 for the minimum number of miRNA-target interactions, 1 for the adjusted p-value (FDR), and with filter by evidence categories set to strong [24]. The miRNA-target genes network was built using the “MIENTURNET” tool [25], while the final network was visualized using Cytoscape software [26]. Functional enrichment analysis of target genes list was performed using the “Enrichr” platform including Kyoto Encyclopedia of Genes and Genomes (KEGG) pathway enrichment analysis. Spearman rank correlation tests were calculated to assess the correlation between blood parameters and the levels of DE miRNAs and snoRNAs employing “ggstatsplot” R package. Other helper packages in plotting and formatting include “dplyr”, “kableExtra”, “gtsummary”, “flextable”, “officer”, “grid”, “VennDiagram”, “pheatmap”, “gplots”, “ggplot2”, “stringr”, “reshape2”, “corrplot”, “tidyr”, “tidyverse”.

Age was normally distributed and hence mean ± standard deviation was reported. Since all clinical variables had a non-normal distribution, comparisons between different groups of severity were performed using the Kruskal–Wallis rank sum test for continuous variables, and Fisher’s Exact Test for categorical. Clinical values were reported as medians and interquartile range [IQR]. Two-tailed p-values were calculated, and *p*-value < 0.05 was considered statistically significant. The R code that was used to generate the figures and analysis is available as a Appendix A.

## 3. Results

First, we identified differentially expressed miRNAs and snoRNAs in severe patients as compared to mild and asymptomatic cases, a |log_2_ (fold change)| > 1.5 and an FDR < 0.05 were used as cutoffs. We found 12 differentially expressed miRNAs and snoRNAs in severe versus asymptomatic, and 31 differentially regulated miRNAs and snoRNAs in severe versus mild cases. Of the 4 miRNAs and snoRNAs that were unique in severe versus asymptomatic, there were 3 upregulated and 1 downregulated transcripts, whereas, of the 23 uniquely differentially expressed miRNAs and snoRNAs in severe versus mild, 7 were upregulated and 16 downregulated. Interestingly, we also observed 3 miRNAs and 5 snoRNAs that were common in both severe versus mild, and severe versus asymptomatic comparisons (Figure 1A). Altogether, differential expression (DE) analysis showed a signature of 27 unique miRNAs and snoRNAs that was associated with COVID-19 severity (Figure 1B,C). Notably, the most highly differentially expressed miRNAs and snoRNAs in severe versus asymptomatic are hsa-miR-1246, ACA40, hsa-miR-4532, hsa-miR-145-5p, and ACA18. Furthermore, the top 5 differentially expressed miRNAs and snoRNAs in severe versus mild are ACA40, hsa-miR-3609, ENSG00000212378 (SNORD78), hsa-miR-1231, hsa-miR-885-3p (Appendix A). Additionally, we report for the first time 9 differentially expressed snoRNAs, namely ACA18 (SNORA18), ACA20 (SNORA20), ACA40 (SNORA40), ACA57 (SNORA57), ENSG00000212378 (SNORD78), U17b, U44, U78, and U79 (Appendix A). Our novel findings suggest that SARS-CoV-2 infection benefits from gene expression programs regulated by snoRNAs, including C/D box snoRNAs (SNORDs) and H/ACA box snoRNAs. Interestingly, no differentially expressed miRNAs or snoRNAs were found in mild versus asymptomatic comparison with the applied cut-offs. These data suggest that both miRNAs and snoRNAs dysregulation is associated with a severe form of COVID-19.

Next, we identified putative targets for the identified differentially expressed miRNAs and snoRNAs in a multi-ethnic cohort of 29 COVID-19 patients residing in Qatar and presenting with varying disease severity. Then we used miRTarBase package validated databases (mirecords, mirtarbase, and tarbase) to identify putative pathways that may be implicated in severe cases of COVID-19. To this end, we used the miRTarBase database, which is curated with >50,000 miRNA-target interactions and validated experimentally by reporter assay, Western blot, microarray, and next-generation sequencing. In total, we found 2584 unique validated target genes for the differentially expressed miRNAs. The resulting target genes list was used in functional enrichment analyses to identify top deregulated pathways in severe COVID-19 patients. Remarkably, key enriched KEGG pathways for the 2584 miRNA-target genes included mainly cancer-related pathways such as pathways in cancer, hepatocellular carcinoma, chronic myeloid leukemia, microRNAs in cancer, breast cancer, gastric cancer, pancreatic cancer, colorectal cancer, proteoglycans in cancer, cellular senescence (Figure 2B). Furthermore, other hallmark gene sets in the Molecular Signatures Database with important biological implications at the top of the list include TNF-alpha Signaling via NF-kB, UV Response Dn, IL-2/STAT5 Signaling, Hypoxia, Inflammatory Response, Apoptosis, PI3K/AKT/mTOR Signaling, G2-M Checkpoint (Appendix A) [27]. Interestingly, miRNA-target gene network analysis (MTGN) elucidated 7 nodes of differentially expressed miRNAs namely hsa-miR-145-5p, hsa-miR-199a-5p, hsa-miR-98-5p, hsa-miR-139-5p, hsa-let-7i-5p, hsa-miR-1246, and hsa-miR-572 (Figure 2A,C). We demonstrate 4 regulated genes targeted by more than 6 differentially expressed miRNAs (Appendix A).

The prospect of using blood miRNAs and snoRNAs as biomarkers can be instrumental in identifying patients with a higher risk of severity and mortality. In this context, we calculated a correlation matrix to examine individual associations between the identified common (Figure 3) and unique (Appendix A) differentially expressed miRNAs and snoRNAs in severe cases, and routine laboratory tests such as complete blood count (CBC), glucose, electrolytes, liver, and kidney function parameters and inflammatory markers. Several of these clinical markers have been linked to prognosis in COVID-19 patients. We found a significant correlation between several differentially expressed miRNAs and snoRNAs and certain hematological and serological parameters, including WBC count (Figure 4A), lymphocyte (%) (Figure 4B), ANC, hematocrit, and albumin (Appendix A). Table 1 represents a summary of patient’s demographics and clinical parameters used in the correlation analysis. These observations can be validated in larger cohorts to help in the early detection of patients at higher risk of severity, and thus in selection of dynamic treatment regimens.

## 4. Discussion

Numerous differentially expressed miRNAs that are observed in this study have been highlighted in host-pathogen interactions [28]. For instance, the family of let-7e/miR-125a/miR-200 miRNAs have been reported to mediate ACE2 gene silencing [29]. Likewise, KEGG enrichment analysis of the differentially expressed miRNAs targets has shown pathways involved in cellular proliferation, invasion, and apoptosis such as PI3K/AKT/mTOR signaling. Whereas, inhibition of hsa-miR-1246 was demonstrated to reduce the cytotoxicity of the Ebola virus glycoprotein in vitro. With multiple viral miRNAs sharing similarities with the host miRNAs, in silico computational studies also uncovered several putative host miRNAs involved in controlling viral replication and limiting disease progression [14,30,31,32]. These host miRNAs are predicted to bind viral sequences modulating cellular immune responses, metabolic pathways, and inhibiting host miRNA maturation, thus facilitating viral escape. Notably, we found 10 differentially expressed miRNAs targeting CDKN1A, a potent cyclin-dependent kinase inhibitor, also known as p21. Interestingly ivermectin inhibits p21-activated kinase 1 (PAK1), a serine/threonine kinase with oncogenic activity. Caly et al. reported that ivermectin decreased SARS-CoV-2 RNA viral load in vitro by 5000-fold with a single treatment [33]. Another target of multiple differentially expressed miRNAs is a suppressor of cytokine signaling 7 (SOCS7), which is known to exert an inhibitory effect on interferon responses, thereby facilitating viral replication. Moreover, insulin-like growth factor I receptor (IGF1R) was also regulated by 6 different differentially expressed miRNAs detected in severely ill patients. IGF1R has been reported as a critical regulator of transformation events, with high levels of expression in most malignancies, where it acts as an anti-apoptotic agent by enhancing cell survival [34]. The MTGN analysis suggests that SARS-CoV-2 infection is regulated by complicated miRNA regulatory networks, through multiple miRNAs targeting the same gene, and single miRNAs targeting multiple genes.

Several studies have reported the fundamental role of interferons in COVID-19 disease severity [35]. Similarly, the identified miRNA in this study (Figure 2) suppresses interferon signaling via binding interferon target genes. Additionally, multiple viruses, including SARS-CoV-2 have been reported to enhance TGF-β signaling, which is known to induce fibrosis and suppress adaptive immunity [36]. Our data suggests a modulation of TGF-β signaling, via the surface receptors and canonical SMAD and MAPK pathways, regulating adaptive immune responses and tissue repair. These findings are in line with the relative lymphopenia reported in severe COVID-19 [1]. Captivatingly, siRNA studies targeting candidate snoRNAs provide an evidence of their functional roles in virus–host interactions against numerous viruses, while knock-down studies have demonstrated that RNA viruses require specific C/D box snoRNAs for optimal replication [17]. Thus, a validated COVID-19 miRNA and snoRNAs signature could be a useful tool to discriminate COVID-19 infections from other respiratory viral infections, identify asymptomatic infections, and develop proteins and metabolite-based tests. This report provides a systematic identification of miRNAs and snoRNAs profile in the blood of SARS-CoV-2 patients with different disease severity, which expands on the previous computational approaches in COVID-19-associated miRNAs and snoRNAs profiling. The patient samples were obtained at a single time point ranging between 2–5 days post COVID-19 diagnosis in Qatar. The patients’ clinical history and lab investigations are well documented in the national healthcare database (CERNER), based on which none of the studied patients were previously diagnosed with any chronic illness, cancer, or immunological disorder, except 18 patients who had type 2 diabetes mellitus. Outcomes from our study can guide future studies involving longitudinal sampling and analysis of circulatory miRNAs/snoRNAs that play a role in disease resolution versus those involved in the development of long COVID. Moreover, future studies with a larger sample size are needed to delineate miRNAs and snoRNAs profiling in females, as well as in children. This study has extended our understanding of the miRNAs and snoRNAs regulatory mechanisms underlying the pathogenesis of SARS-CoV-2 infection, which can be explored further to produce promising therapies.

## Figures and Tables

**Figure 1 vaccines-09-01056-f001:**
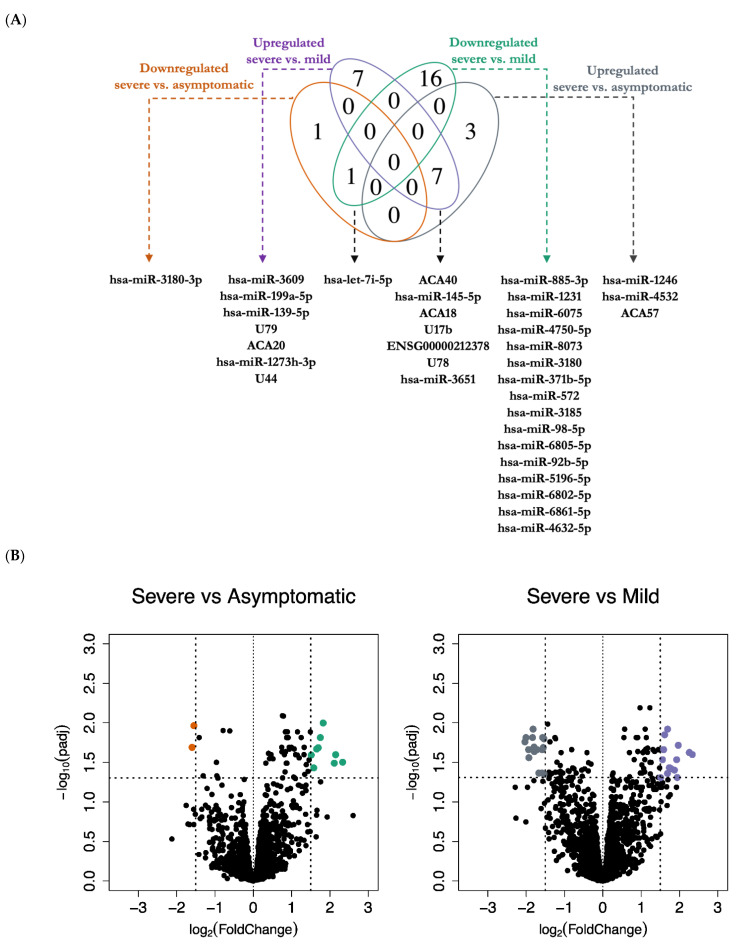
Differential expression analysis of miRNAs and snoRNAs in asymptomatic, mild, and severe COVID-19 patients. (**A**) A Venn diagram of the number of unique and common differentially expressed miRNAs and snoRNAs in severe versus asymptomatic (D: grey: upregulated, A: orange: downregulated), and in severe versus mild (C: purple: upregulated, B: green: downregulated). The overlap of the two ovals with upregulated transcripts represents 7 miRNAs and snoRNAs shared between the two upregulated lists, and the two ovals with downregulated transcripts represent 1 miRNA shared between the two downregulated lists. The areas of no overlap represent the number of uniquely (not shared) miRNAs/snoRNAs in each respective comparison, namely 4 in severe versus asymptomatic, and 23 miRNAs/snoRNAs in severe versus mild comparison. (**B**) Volcano plots of differentially expressed miRNAs and snoRNAs, with y-axis as-log_10_ adjusted *p*-value (adjusted for multiple testing with the Benjamini–Hochberg procedure) and x-axis as the log_2_-fold change. The vertical lines represent a threshold of |log_2_-fold change| > 1.5, either upregulated (right side) or downregulated (left side), while the horizontal lines represent an FDR < 0.05. Each colored point represents an individual probe in severe versus asymptomatic comparison (D: green: upregulated, A: orange: downregulated), and in severe versus mild comparison (C: purple: upregulated, B: grey: downregulated). (**C**) Hierarchical clustering of Log_2_ normalized expression values of the differentially expressed miRNAs and snoRNAs in severe versus asymptomatic comparison (n = 25), and in severe versus mild (n = 50).

**Figure 2 vaccines-09-01056-f002:**
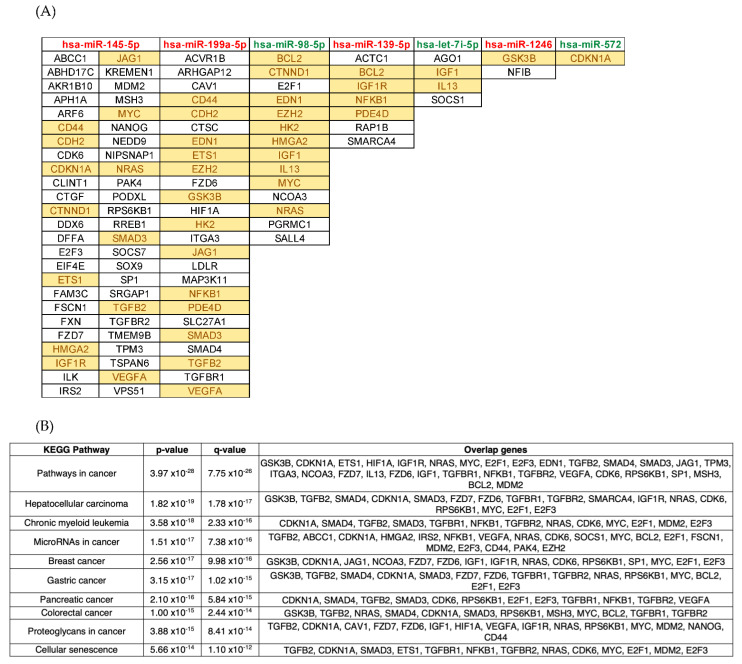
Functional and pathway enrichment analyses using a list of common differentially expressed miRNAs shared by severe versus asymptomatic, and severe versus mild comparisons. (**A**) Representation of differentially expressed miRNAs and their associated target genes. The name of the miRNA is highlighted in red for upregulated, and in green for downregulated, whereas target genes that are highlighted in yellow are those that are targeted by more than 1 miRNA. (**B**) KEGG enrichment analysis with the names of top 10 significant deregulated pathways for KEGG 2021 Human database. The q-value is an adjusted *p*-value calculated using the Benjamini–Hochberg method for correction for multiple hypotheses testing. (**C**) Network-based visualization of differentially expressed miRNAs and their associated target genes, where red/green nodes represent the upregulated/downregulated miRNA, and the blue nodes represent its targeted gene.

**Figure 3 vaccines-09-01056-f003:**
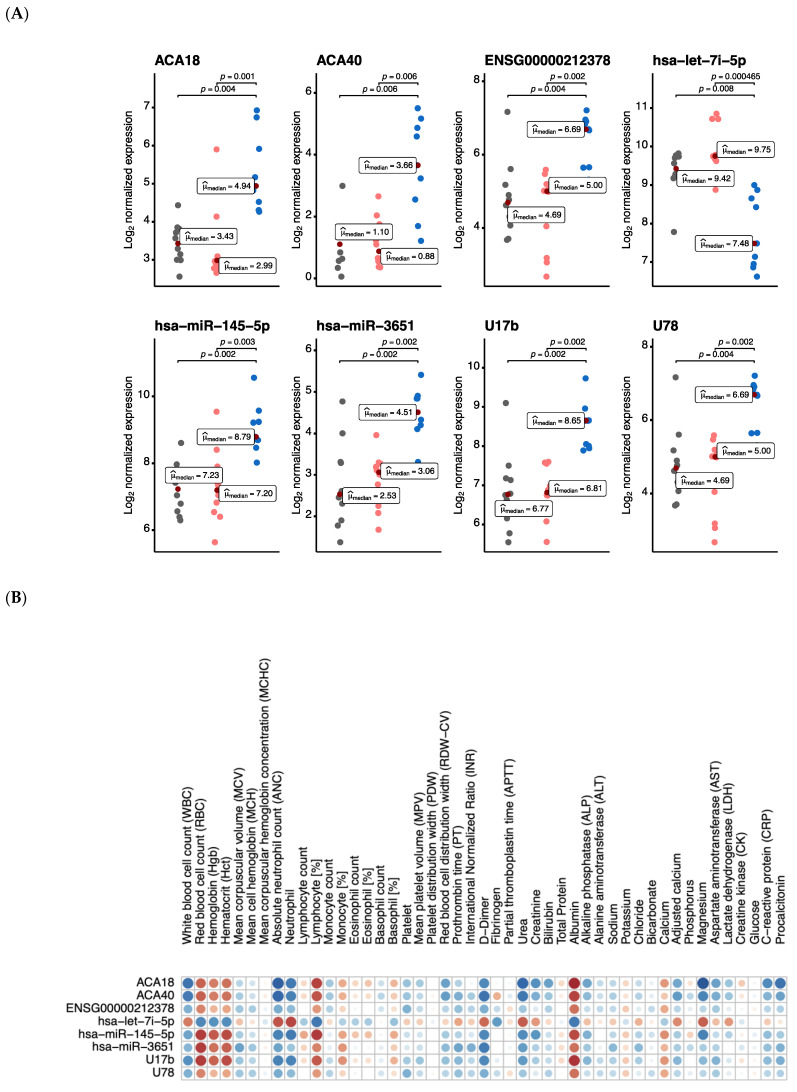
Association of routine clinical markers with differentially expressed common miRNAs and snoRNAs. (**A**) Scatter plots of Log_2_ normalized expression values of miRNAs and snoRNAs in asymptomatic (n = 10, grey), mild (n = 10, light orange), and severe (n = 9, blue) cases. The comparison between different severity groups was done using the non-parametric Kruskal–Wallis one-way ANOVA test, the median for asymptomatic, mild, and severe groups is reported as a measure of centrality in each scatter plot. In each respective comparison, the p-adjusted values were calculated using Holm’s sequential Bonferroni procedure for multiple hypothesis tests at an alpha level of 0.05. (**B**) A correlation matrix of the 8 miRNAs and snoRNAs with associated clinical markers. Each cell contains a correlation coefficient between the possible pairs of variables, namely Spearman’s rho statistic calculated at a significance level of 0.05. Color scale ranges from blue (r = −1) to white (r = 0) to red (r = 1).

**Figure 4 vaccines-09-01056-f004:**
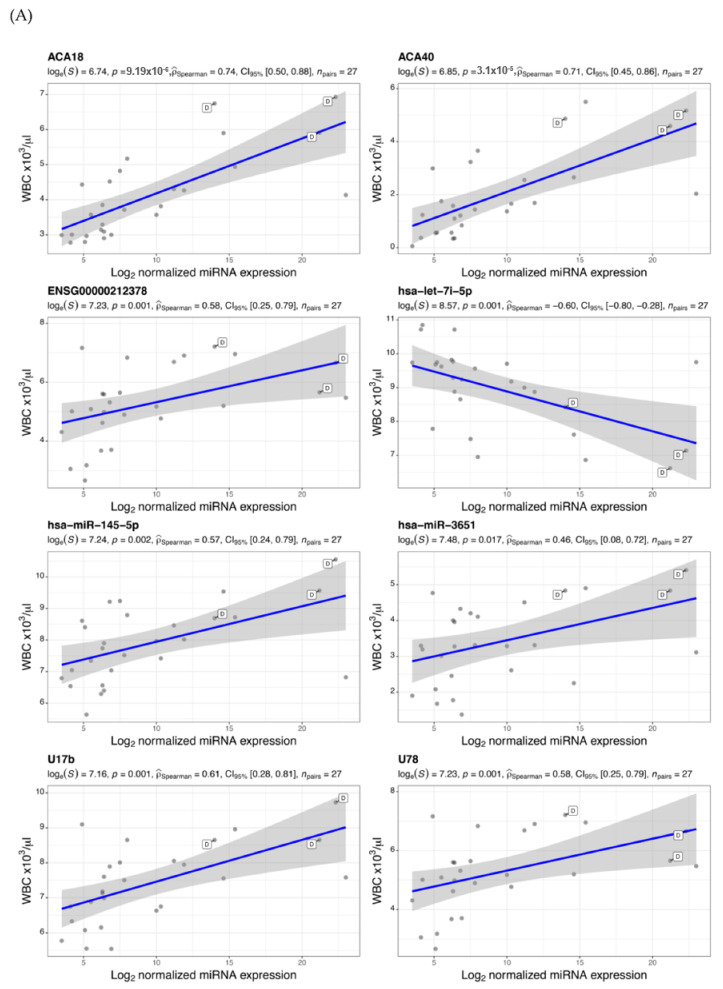
Correlation of (**A**) WBC count and (**B**) lymphocytes (%) with levels of common differentially expressed miRNAs and snoRNAs. Scatter plots x-axis represents Log_2_ normalized miRNA expression level and y-axis as the clinical marker measurement. Each dot represents a single miRNA transcript, and those tagged with D letter show a deceased patient measurement. For all statistical tests in the plots, the APA standard for statistical reporting is shown for Spearman correlation test including evidence in favor of null over alternative hypothesis, natural logarithm of Bayes Factor, *p*-value, Spearman’s rank correlation coefficient, confidence intervals, and number of observations.

**Table 1 vaccines-09-01056-t001:** Clinical characterization of asymptomatic, mild, and severe COVID-19 patients. Patients were grouped based on clinical disease severity including severe requiring oxygen support and ICU admission, mild cases were recognized as those with clinical symptoms and positive radiographic findings suggestive of pulmonary involvement. * ALP: Alkaline phosphatase, ALT: Alanine aminotransferase, ANC: Absolute neutrophil count, APTT: Partial thromboplastin time, AST: Aspartate aminotransferase, CK: Creatine kinase, CRP: C-reactive protein, Hct: Hematocrit, Hgb: Hemoglobin, IL-6: Interleukin-6, INR: International normalized ratio, LDH: Lactate dehydrogenase, MCH: Mean cell hemoglobin, MCHC: Mean corpuscular hemoglobin concentration, MCV: Mean corpuscular volume, MPV: Mean platelet volume, PDW: Platelet distribution width, PT: Prothrombin time, RBC: Red blood cell count, RDW-CV: Red blood cell distribution width, WBC: White blood cell count. Values are represented as: n (%), median (25–75%). For age: Mean ± SD is reported. The statistical tests used to calculate *p*-value are Fisher’s exact test or Kruskal–Wallis rank sum test.

Characteristic *	N	Asymptomatic	Mild	Severe	*p*-Value
Age [years]	29	54.50 ± 4.58	49.60 ± 9.18	58.22 ± 12.88	0.078
COVID-19 average CT	21	22.9 (21.5–29.2)	26.7 (23.4–29.4)	33.8 (26.8–34.1)	0.3
Diabetes Mellitus	29	6 (20.68%)	6 (20.68%)	6 (20.68%)	>0.9
Glucose [mmol/L]	20	7.00 (6.25–14.40)	6.80 (6.27–8.12)	6.60 (6.40–10.60)	0.9
WBC [×10^3^/μL]	27	6.3 (6.2–7.8)	5.5 (5.1–6.4)	11.9 (8.0–15.4)	0.014
Lymphocyte count [×10^3^/μL]	27	2.10 (1.20–2.50)	1.30 (1.20–1.61)	1.10 (0.90–1.40)	0.2
Lymphocyte [%]	27	29 (24–33)	24 (21–31)	9 (5–14)	0.004
ANC [×10^3^/μL]	27	3.6 (2.9–6.1)	3.6 (2.7–4.7)	7.7 (6.7–13.1)	0.005
Neutrophil [%]	21	56 (51–62)	66 (58–69)	83 (80–86)	0.012
Eosinophil count [×10^3^/μL]	27	0.10 (0.00–0.27)	0.00 (0.00–0.00)	0.00 (0.00–0.10)	0.2
Eosinophil [%]	27	2.50 (0.90–3.40)	0.10 (0.00–0.80)	0.10 (0.00–0.80)	0.029
Monocyte count [×10^3^/μL]	27	0.60 (0.60–0.70)	0.47 (0.40–0.50)	0.90 (0.80–1.00)	0.11
Monocyte [%]	27	8.70 (6.80–10.90)	9.00 (5.10–9.40)	5.60 (4.80–6.60)	0.14
Basophil count [×10^3^/μL]	27	0.030 (0.030–0.040)	0.020 (0.020–0.050)	0.030 (0.010–0.040)	0.6
Basophil [%]	27	0.50 (0.40–0.80)	0.40 (0.20–0.50)	0.20 (0.10–0.40)	0.026
RBC [×10^6^/μL]	27	5.10 (4.90–5.30)	5.00 (4.70–5.70)	3.80 (3.50–4.20)	0.011
Hgb [g/dL]	27	13.60 (13.50–15.20)	14.20 (13.60–16.00)	11.20 (9.90–12.60)	0.036
HbA1C [%]	18	8.70 (6.05–11.80)	7.00 (6.40–7.50)	6.00 (5.90–6.40)	0.4
Hct [%]	27	42 (40–43)	42 (40–47)	33 (30–38)	0.03
MCV [fL]	27	81.9 (81.0–87.0)	85.4 (82.2–91.2)	90.6 (86.4–91.2)	0.031
MCH [pg]	27	27.30 (26.50–29.50)	28.10 (27.90–30.50)	30.00 (29.40–30.30)	0.11
MCHC [g/dL]	27	33.30 (32.40–34.00)	33.60 (32.90–34.10)	33.50 (32.50–34.10)	0.8
RDW-CV [%]	27	13.00 (11.90–14.30)	14.20 (12.70–14.90)	13.90 (12.50–16.20)	0.3
MPV [fl]	27	10.20 (9.60–11.30)	10.80 (9.90–12.00)	10.40 (9.80–11.50)	0.8
Platelet [×10^9^/L]	27	226 (200–251)	187 (161–324)	342 (223–363)	0.2
Ferritin [μg/L]	18	574 (574–574)	449 (210–804)	1131 (700–1856)	0.079
PDW [fl]	19	15.10 (15.10–15.10)	13.90 (12.00–15.80)	11.70 (11.00–14.20)	0.5
Fibrinogen [g/L]	11	6.80 (6.80–6.80)	4.60 (4.60–4.60)	5.10 (3.30–5.60)	0.5
D-Dimer [mg/L FEU]	16	1.55 (1.55–1.55)	0.40 (0.32–0.60)	2.65 (2.15–5.16)	0.036
APTT [second]	15	26 (26–26)	32 (29–35)	31 (31–36)	0.4
PT [second]	15	12.60 (12.60–12.60)	11.30 (11.10–11.50)	13.10 (12.60–13.50)	0.047
INR	15	1.10 (1.10–1.10)	1.00 (1.00–1.00)	1.10 (1.10–1.10)	0.057
CRP [mg/L]	27	3 (2–6)	21 (5–56)	26 (5–78)	0.063
IL-6 [pg/mL]	9	-	32 (20–34)	306 (137–476)	0.2
Total protein [g/L]	24	76 (66–77)	73 (69–75)	71 (68–72)	0.8
Albumin [g/L]	26	40 (36–42)	37 (35–39)	26 (24–29)	0.001
Chloride [mmol/L]	25	100.0 (97.5–101.5)	100.7 (98.0–103.0)	104.0 (102.0–107.0)	0.079
Magnesium [mmol/L]	14	0.85 (0.85–0.85)	0.79 (0.76–1.00)	0.94 (0.89–1.09)	0.4
Potassium [mmol/L]	26	4.80 (4.55–5.05)	4.10 (3.80–4.40)	4.20 (3.60–4.60)	0.11
Sodium [mmol/L]	27	136 (134–138)	138 (136–140)	139 (137–148)	0.2
Bicarbonate [mmol/L]	23	25.0 (24.0–26.0)	26.0 (21.5–27.0)	23.0 (22.0–28.0)	>0.9
Calcium [mmol/L]	26	2.32 (2.25–2.37)	2.38 (2.34–2.44)	2.19 (2.14–2.30)	0.093
Adjusted calcium [mmol/L]	26	2.29 (2.24–2.39)	2.46 (2.35–2.46)	2.46 (2.45–2.52)	0.039
Vitamin D [ng/mL]	14	23 (20–24)	23 (14–32)	32 (23–39)	0.6
Procalcitonin [ng/mL]	13	0.25 (0.25–0.25)	0.38 (0.22–0.38)	0.26 (0.10–0.72)	>0.9
Bilirubin [mg/dL]	25	5 (5–16)	9 (5–12)	10 (7–30)	0.4
Urea [mmol/L]	27	4 (4–6)	4 (4–5)	16 (10–20)	<0.001
Uric acid [µmol/L]	19	295 (282–311)	229 (210–270)	355 (337–374)	0.2
Creatinine [µmol/L]	27	85 (73–99)	78 (69–100)	115 (83–168)	0.3
ALT [U/L]	23	18 (17–25)	34 (20–55)	59 (20–134)	0.2
AST [U/L]	20	23 (18–33)	27 (24–34)	34 (28–110)	0.2
CK [U/L]	14	129 (129–129)	97 (62–173)	103 (62–239)	0.8
ALP [U/L]	25	79 (66–115)	66 (61–93)	78 (45–143)	0.9
LDH [U/L]	14	183 (183–183)	337 (226–343)	332 (294–375)	0.3

## Data Availability

Additional data and code generated during the current study are available at EMBL-EBI ArrayExpress (https://www.ebi.ac.uk/fg/annotare/edit/13429/) under ArrayExpress accession E-MTAB-10970.

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
