# Peer review of "SnoRNAs and miRNAs Networks Underlying COVID-19 Disease Severity"

_vaccines, 2021, doi:10.3390/vaccines9101056_

Round 1

Reviewer 1 Report

In their manuscript, Parray et al. analyze a small cohort of SARS-CoV-2 positive male patients in Qatar with different COVID-19 severities (asymptomatic, mild, severe; ~10 patients/group). They use microarray to quantify miRNA and snoRNA expression in peripheral blood, identifying differentially expressed (DE) miRNAs and snoRNAs. They then use public databases to get targets of DE miRNAs and perform pathway analysis to implicate pathways that may be important for COVID-19 severity. Finally, they attempt to correlate clinical markers with DE miRNA/snoRNA expression.

While the premise of the study is fine, there are egregious issues with the manuscript and this reviewer recommends the article be rejected. The authors oversell the study by failing to discuss its limitations and using broad terms like “ncRNAs” to discuss miRNAs and snoRNAs. Additionally, this reviewer questions the rigor of the bioinformatic analysis used to identify DE miRNAs/snoRNAs because 1) the data/code are not deposited in the required public repositories and 2) DE analysis was done with an extremely lenient FDR of <0.2. Furthermore, the manuscript is devoid of appropriate citations for the packages and databases used in the bioinformatic analysis; this is not acceptable and borders on unethical. The manuscript also contained many small grammatical errors, and certain figures (i.e. Figure 4) contain obvious errors that should be easily caught by the authors. Unfortunately, after reviewing Figure 4, this reviewer deemed it inappropriate to spend their time giving comments if the authors cannot proof their manuscript, and recommends the article be rejected. Examples of major and minor issues are provided below – the issues with this manuscript are too numerous to provide an exhaustive list.

Major comments

Overselling:

  • Define “ncRNAs” explicitly in introduction. This is a broad term, not limited to miRNAs and snoRNAs. What about tRNAs, lncRNAs, siRNAs ? The use of the “ncRNAs” to describe only miRNAs and snoRNAs throughout the manuscript is misleading.
  • Line 62: “There are no reports yet with a comprehensive characterization of ncRNAs in SARS-CoV-2 infection, therefore, we report for the first time the snoRNA and miRNA signature in the peripheral blood of severe COVID-19 cases (n=9), as compared to mild (n=10) and asymptomatic(n=10) patients.” This is not a “comprehensive study” of “ncRNAs” (see bullet above). It would be more accurate to say it is the first study to include differential expression of snoRNAs during COVID-19 disease (by now there are numerous studies looking at miRNAs during COVID-19 disease in a similar context, some of which are cited in this paper).
  • Lines 174-176: “Next, we searched for putative targets for all identified DEMIs in the blood of severely ill COVID-19 patients, and then constructed a network of miRNA targets to reveal pathways and mechanisms underpinning disease severity.” Please do not over-interpret results – from a cohort of ~10 male patients/group from one geographic region, some DE miRNAs are identified. Then, the authors use miRTarBase (not cited) to get miRNA targets and find enriched gene sets. The authors did not “construct a network of miRNA targets to reveal pathways and mechanisms”, they used a database relying on the work of many prior researchers to get miRNA targets without citing the database properly, then did pathway analysis and looked for commonly regulated targets; the results at best implicate putative pathways, but not mechanisms.

The authors fail to discuss the limitations of their study: all male, small sample size, all from one geographic region, potential immunodeficiencies of patients, timing of diagnosis/infection/disease onset compared to sample collection.

  • Can the authors justify why only male patients were included? The reviewer mentions this because sex-specific differences in COVID-19 severity have been previously shown.
  • Do the authors have genetic information from patients? If so, this would be important to mention as underlying genetic disorders can alter COVID disease severity and/or miRNA signatures. Would it not make sense to exclude patients with well-characterized immune mutations (e. IFNAR SNPs); or least discuss this possibility, lacking this information? In contrast to the author’s statement that “co-morbidities” were matched, this reviewer only saw diabetes noted in Table S6 (which was not referenced in the main text).
  • The authors say that the patients were age-matched – so, does this mean that the distribution of ages was the same for each group? Or, is there a skew in the ages of severe versus mild versus asymptomatic cases? This reviewer asks because miRNA signatures may also change for different age groups. This information in Table S6 was confusing and lacked appropriate explanation for many of the numbers shown. g., how can “Age [years]” of the asymptomatic patients be 54 (52-56) – is 54 the average and 52-56 the range? This is not stated. Another example: for Diabetes Mellitus, do 6 patients in each group have diabetes? Why is the percentage for all groups the same if the number of patients per group is different (10 asymptomatic, 10 mild, 9 severe)? If 6/29 patients overall have diabetes, where does 33% come from.
  • Finally, were the blood samples used for analysis taken at any specific time interval following identification of the patients as SARS-CoV-2 positive? Couldn’t timing post-diagnosis/post-infection and/or disease onset make quite a difference in miRNA/snoRNA expression profiles?

Information/rigor in bioinformatic analysis is lacking:

  • Is the code used for the analysis publicly available (e. in GitHub or supplementary data)? This is required by the journal guidelines, and this reviewer would not be able to reproduce the authors’ analysis with the information given in the methods.
  • Why was a log2FC of 1.5 chosen as the cutoff for differentially expressed miRNAs/snoRNAs, can the authors justify? How does this cutoff compare to other papers looking at differential miRNA/snoRNA expression? FDR<0.2 is very lenient, the standard is ~0.05; why was this chosen?
  • Figure 2. How were the mentioned KEGG pathways chosen? All pathways reported, p-value cutoff, top 10 unique enriched pathways per group? Also, please cite MSigDB appropriately.

Are the expression data deposited in an appropriate public repository (i.e. GEO, ArrayExpress?) This is required by the journal guidelines.

Packages/software used for analysis must be cited in the references.

Can the authors do network analysis on the snoRNAs that were differentially expressed in different group comparisons? There is not too much focus on the snoRNAs in the results.

Figure 4. This figure is negligent and shows utter lack of proofing, as the authors only show 3 points in these correlations, all from deceased patients. This reviewer assumes Figure 4 should look similar to Figure S2.

Minor comments

By no means comprehensive, please take the time to proofread your manuscript!

Lines 22-25: suggest “Using differential expression analysis of microarray data (n=29), we identified hsa-miR-494-3p, hsa-miR-1246, ACA40, hsa-miR-4532, and ACA15 as the top five most significant differentially expressed ncRNAs transcripts in severe versus asymptomatic cases;, and ACA40, hsa-miR-3609, hsa-miR-24 6790-5p, hsa-miR-126-3p, and hsa-miR-885-3p were the five most significant differentially expressed ncRNAs five in severe versus mild cases.”

Define WBC, RBC the first time they appear.

Line 56: “is important” - would suggest may be important, this is what the authors are trying to show, right?

Line 57-59: grammar – multiple interactions, right?

Line 66: this sentence should be edited for clarity and probably belongs in the methods.

Line 77: typo? “Form” should be “from”?

Line 138: it’s a little strange to define differentially expressed miRNAs as “DEMIs” but then use this in combination with snoRNAs (as in line 150 “the top 5 DEMIs/snoRNAs”)... the snoRNAs are differentially expressed as well, right?

Line 151: remove the period after “Table.” Idem for all subsequent references to Tables in the text.

Line 158-159: no DEMIs were found in mild versus asymptomatic COVID-19 cases – what about differentially expressed snoRNAs in mild versus asymptomatic COVID-19 cases?

Figure 1A: it would be easier if the Venn diagram labels would be changed from “A”, “B”, “C”, “D” to descriptive labels.

Figure 1B legend, line 168: define “padj” explicitly in legend.

Figure 1C: heatmap legend appears cutoff in my PDF. So, the left heatmap is severe versus asymptomatic, and the right heatmap is severe versus mild? Can the authors specify this in the legend?

Figure 2: Can the authors label the miRNAs in the blue nodes? Can sets of target genes implicated in KEGG analyses be highlighted in the MTGN analysis (yellow nodes)? The figure would benefit from labeling different panels.

Figure 3: It seems figure legend for panels A and B are reversed.

Reviewer 2 Report

This research article by Aijaz Parray et al. studies the miRNA and snoRNA profile in asymptomatic, mild, and severe COVID-19 patients to identify potential COVID-19 associated biomarkers. The study collected blood samples from patients diagnosed with COVID-19 and used microarray technology as the miRNA expression analysis. Several data interpretations including differential expression ncRNA analysis, functional and pathway enrichment analysis, and correlation analysis with clinical markers, were performed. The study identified several significant ncRNAs signatures for COVID-19 cases with different disease severity. Although the findings in this research highlight the importance of ncRNAs as biomarkers in SARS-CoV-2 infection, the article needs to be further improved before publication. Issues in this research article are listed as below.

  1. It’s important, and would be clearer if the authors can list the COVID-19 positive participants information in a table including the number, gender, and age etc.
  2. For microarray analysis, healthy people that were COVID-19 negative should be included for comparison with positive patients. Recognizing the clusters of miRNA/snRNA differently expressed between COVID-19 positive patients and healthy people is critical before comparing those with different disease severity.
  3. The statistical analysis method should be clarified in figure3&4.
  4. The findings demonstrate the significance of the ncRNA profile associated with COVID-19 diagnosis, however, the methodology used in this study is limited. The author should at least provide their perspectives on how to improve the experimental design, methodological strategy, and data analyses to strictly control the robustness and reproducibility of their finding. This is important for its translation to clinical practice.

Author Response

Dear Editor,

Thank you very much for your interest in our work. We are delighted by the decision that we may prepare a revision of our above-mentioned manuscript. We gratefully thank the reviewers for their extremely valuable comments and helpful feedback that guided us to improve our manuscript significantly, and greatly appreciate the thorough and thoughtful comments provided. We made sure that each one of the reviewer comments has been addressed carefully and the paper is revised accordingly. In the following, you will find a point-by-point reply to the comments of the reviewers. The comments are shown in black and our responses in red. We will be happy to address any further points, if you still have any questions or concerns about the manuscript.

point 1: This research article by Aijaz Parray et al. studies the miRNA and snoRNA profile in asymptomatic, mild, and severe COVID-19 patients to identify potential COVID-19 associated biomarkers. The study collected blood samples from patients diagnosed with COVID-19 and used microarray technology as the miRNA expression analysis. Several data interpretations including differential expression ncRNA analysis, functional and pathway enrichment analysis, and correlation analysis with clinical markers, were performed. The study identified several significant ncRNAs signatures for COVID-19 cases with different disease severity. Although the findings in this research highlight the importance of ncRNAs as biomarkers in SARS-CoV-2 infection, the article needs to be further improved before publication. Issues in this research article are listed as below.

Response 1: We thank the reviewer for taking the time to assess our manuscript.

Point 2: It’s important, and would be clearer if the authors can list the COVID-19 positive participants information in a table including the number, gender, and age etc.

Response 2: We thank the reviewer for this suggestion and present this information as a main table (Table 1) in the manuscript instead of supplementary table 6.

Point 3: For microarray analysis, healthy people that were COVID-19 negative should be included for comparison with positive patients. Recognizing the clusters of miRNA/snRNA differently expressed between COVID-19 positive patients and healthy people is critical before comparing those with different disease severity.

Response 3: We thank the reviewer for this insightful comment. Although we tend to agree, the scope of our research was the comparison of disease-severity within severe, mild and asymptomatic COVID-19 patients. The aim is to identify patients who are at high risk of developing severe disease associated with low survival using molecular targets such as miRNAs and snoRNAs, and to correlate these ncRNAs with available clinical markers. This is especially important with the increased need to identify biomarkers to predict disease severity and mortality during the earlier stages of COVID-19, which can aid in treatment, developing new management strategies, and allocation of resources to improve survival. In addition to potentially assisting clinicians in early recognition of patients at risk for critical complications. We duly acknowledge the importance of using healthy control subjects to address the functional role of these identified differentially expressed miRNAs and snoRNAs in COVID-
19 disease compared to non-disease status.

Point 4: The statistical analysis method should be clarified in figure3&4.

Response 4: We thank the reviewer for raising this point and apologize for the lack of statistical information. We have added the following statements to the legend of figure 4 “For all statistical tests in the plots, the APA standard for statistical reporting is shown for spearman correlation test including evidence in favor of null over alternative hypothesis, natural logarithm of Bayes Factor, p-value, Spearman's rank correlation coefficient, confidence intervals, and number of observations”. Furthermore, the legend of figure 3 now contains
additional information for statistical analysis “The comparison between different severity groups was done using the non-parametric Kruskal–Wallis one-way ANOVA test, the median for asymptomatic, mild and severe groups is reported as a measure of centrality in each scatter plot. In each respective comparison, the p-adjusted values were calculated using Holm’s sequential Bonferroni procedure for multiple hypothesis tests at an alpha level of 0.05. (B) A correlation matrix of the 8 miRNAs and snoRNAs with associated clinical markers. Each cell contains a correlation coefficient between the possible pairs of variables, namely Spearman's rho statistic calculated at a significance level of 0.05. Color scale ranges from blue (r = −1) to white (r = 0) to red (r = 1).”

Point 5: The findings demonstrate the significance of the ncRNA profile associated with COVID-19 diagnosis, however, the methodology used in this study is limited. The author should at least provide their perspectives on how to improve the experimental design, methodological strategy, and data analyses to strictly control the robustness and reproducibility of their finding. This is important for its translation to clinical practice.

Response 5: The reviewer has made excellent points and we sincerely appreciate these well-thought comments, which helped us to improve the manuscript significantly. We have extensively reviewed the methodology and used a more stringent approach for differential expression analysis employing an FDR less than 0.05 to control for robustness. We now present new figures and tables based on the new analysis throughout the revised manuscript. With respect to reproducibility, we have revised the references list and cited all the main packages and tools that we used in the analysis including “oligoCalsses”, “BioBase”, “oligo”, “limma”, “multiMir” and “miRBaseConverter”, “miRTarBase”, “Enricher”. Other used helper packages in formatting and plotting have been mentioned in the methodology section as well, including “dplyr”, “kableExtra”, “gtsummary”, “flextable”, “officer”, “grid”, “VennDiagram”, “pheatmap”, “gplots”, “ggplot2”, “ggstatsplot”, “stringr”, “reshape2”, “corrplot”, “tidyr”, “tidyverse”.

In addition, we now share the full code that was used in analysis and quality control testing of the raw microarrays data, as well as in generating the figures and tables in differential expression analysis. Moreover, the original data are attached as tables in supplementary files to the current version of the manuscript.

A table that includes the available clinical and detailed demographic data with the array number annotation for each patient.

A table that incudes the probes annotation downloaded from affymetrix website [http://www.affymetrix.com/support/technical/byproduct.affx?product=miRNAGalaxy. File name: miRNA-4_0 Annotations, CSV format (23 MB, 10/3/16)] for the used Affymetrix GeneChip 4.0.

We have prepared a submission folder to be deposited in the Gene Expression Omnibus (GEO) database. The data will be deposited as raw CEL files following the confirmation of manuscript acceptance. We reserve this right to wait for making the data public as this is the first to our knowledge snoRNA data in COVID-19 cases. We are willing to provide the reviewer with the 29 CEL files upon request. Noteworthy, our study cohort consisted of adult male subjects only, due to limited sample accessibility that was restricted to a quarantine facility that accepts asymptomatic males only. Additionally, we acknowledge the limited sample size (approximately 10 per group) that warrants the need to expand to larger sample sizes in multi-ethnic populations to validate the findings of our study. Also, despite of the well-documented clinical data, we cannot completely rule out underlying silent pathologies when dealing with human subjects. It is noteworthy to mention that the cohort was analysed based on presence or absence of Type 2 diabetes mellitus instead of severity which yielded no significant targets for the differential expression analysis.
Furthermore, we do not have genetic or sequencing data for this cohort of patients,
which can shed light on the host genetic background, which in turn does not allow us to exclude patients with existing SNPs or immune mutations that may influence disease progression and fatality. Therefore, we have also added in the discussion section the study limitations and raised some points to improve the study design in future studies as follows:

The patient samples were obtained at a single time point ranging between 2-5 days post COVID-19 diagnosis in Qatar. The patients’ clinical history and lab investigations are well documented in the national healthcare database (CERNER), based on which none of the studied patients were previously diagnosed with any chronic illness, cancer, or immunological disorder, except 18 patients who had type 2 diabetes mellitus. Outcomes from our study can guide future studies involving longitudinal sampling and analysis of circulatory miRNAs/snoRNAs that play a role in disease resolution versus those involved in the development of long Covid. Moreover, future studies with a larger sample size are needed to delineate miRNAs and snoRNAs profiling in females, as well as in children.

We appreciate all the valuable comments from the reviewers of our study. We have revised our manuscript, according to the reviewers’ comments, questions, and suggestions. We believe that the revised manuscript has been further improved.

Round 2

Reviewer 1 Report

The revised manuscript by Parray et al. is improved by discussing the limitations of the study, fixing figures and addressing some issues with the bioinformatic analysis. However, this reviewer feels the authors still need to address some points prior to acceptance. Although I note that the authors have prepared a GEO submission with their raw microarray data, the journal does require the data be deposited prior to submission of your manuscript. I understand the author’s concerns about making the data public before acceptance of the manuscript, but you can submit to GEO to provide accession numbers in your manuscript and still control when the data is released publicly. You should do this. I guess enforcing this policy is at the discretion of the editor, but obviously in this reviewer’s eyes, submitting your raw data is important for reproducibility and transparency in science. Also, some of the citations you have added have not been included in your references section, so the actual paper is still not properly cited: e.g. miRTarBase, I assume this is the reference in line 134 of the revised MS “{Huang, 2020 #11}”.  In addition, Table S6 still seems to be unreferenced in the text and you only list Tables S1-S5 (line 325). I also cannot find the code you state (point 19 and line 149) you added to the supplementary material. These issues should be addressed before the paper can be accepted.   

Author Response

Dear Editor, 

We are thankful to the reviewer who has guided us to improve the quality of our manuscript. 

Please see the attachment for a point-to-point response to the reviewers comments.

Reviewer 2 Report

The manuscript by Aijaz Parray et al is improved after revision according to the reviewers’ comments. The authors need to check carefully the article again to correct any grammatical errors. For example, line 42, incorrect spelling 'ageing'. The reviewer will support the publication if these minor issues are revised.

Author Response

Dear Editor, 

We thank the second reviewer for the beneficial comments that have helped us to improve the quality of the manuscript significantly. 

Point 1: The manuscript by Aijaz Parray et al is improved after revision according to the reviewers’ comments. The authors need to check carefully the article again to correct any grammatical errors. For example, line 42, incorrect spelling 'ageing'. The reviewer will support the publication if these minor issues are revised.

Response 1: We would like to thank the reviewer for taking the time and effort necessary to review the manuscript, and we appreciate the positive feedback. As pointed out, “ageing” has been corrected as well as other minor spelling and grammatical errors in the manuscript.

We look forward to hearing from you regarding our revised submission, and to respond to any further questions and comments you may have.
Sincerely,